# Thunderstruck: The ACDC model of flexible sequences and rhythms in recurrent neural circuits

**Cristian Buc Calderon**[1,2,3]*, **Tom Verguts**[2], **Michael J. Frank**[1,3]

**1** Department of Cognitive, Linguistic & Psychological Sciences, Brown University, Providence, Rhode Island, United States of America, **2** Department of Experimental Psychology, Ghent University, Ghent, Belgium, **3** Carney Institute for Brain Science, Brown University, Providence, Rhode Island, United States of America

* cbuccald@gmail.com

**Data Availability Statement:** Code for all simulations is available from https://github.com/CristianBucCalderon/ACDC.

**Funding:** o C.B.C is supported by FWO grant #1207719N. T.V. is supported by BOF17-GOA-

## Abstract

Adaptive sequential behavior is a hallmark of human cognition. In particular, humans can learn to produce precise spatiotemporal sequences given a certain context. For instance, musicians can not only reproduce learned action sequences in a context-dependent manner, they can also quickly and flexibly reapply them in any desired tempo or rhythm without overwriting previous learning. Existing neural network models fail to account for these properties. We argue that this limitation emerges from the fact that sequence information (i.e., the position of the action) and timing (i.e., the moment of response execution) are typically stored in the same neural network weights. Here, we augment a biologically plausible recurrent neural network of cortical dynamics to include a basal ganglia-thalamic module which uses reinforcement learning to dynamically modulate action. This "associative cluster-dependent chain" (ACDC) model modularly stores sequence and timing information in distinct loci of the network. This feature increases computational power and allows ACDC to display a wide range of temporal properties (e.g., multiple sequences, temporal shifting, rescaling, and compositionality), while still accounting for several behavioral and neurophysiological empirical observations. Finally, we apply this ACDC network to show how it can learn the famous "Thunderstruck" song intro and then flexibly play it in a "bossa nova" rhythm without further training.

## Author summary

How do humans flexibly adapt action sequences? For instance, musicians can learn a song and quickly speed up or slow down the tempo, or even play the song following a completely different rhythm (e.g., a rock song using a bossa nova rhythm). In this work, we build a biologically plausible network of cortico-basal ganglia interactions that explains how this temporal flexibility may emerge in the brain. Crucially, our model factorizes sequence order and action timing, respectively represented in cortical and basal ganglia dynamics. This factorization allows full temporal flexibility, i.e. the timing of a learned

004. M.J.F. is supported by NIMH R01MH084840-08A1. C.B.C received a salary from FWO (grant #12O7719N). The funders had no role in study design, data collection and analysis, decision to publish, or preparation of the manuscript.

**Competing interests:** The authors have declared that no competing interests exist.

action sequence can be recomposed without interfering with the order of the sequence. As such, our model is capable of learning asynchronous action sequences, and flexibly shift, rescale, and recompose them, while accounting for biological data.

## Introduction

Learning and manipulating sequential patterns of motor output are essential for virtually all domains of human behavior. For instance, musicians can learn multiple precise spatiotemporal sequences, each with their own rhythm. They can later modify the tempo to a learned sequence, or even apply a completely different rhythm, e.g., perform a rock song with a bossa nova rhythm. Thus, musicians can quickly and flexibly manipulate action timing in action sequences. Similar capabilities abound in other domains, such as language production and athletics.

Precisely timed action sequences are thought to emerge from dynamical neural patterns of activity. In particular, sparse sequential activity patterns observed in basal ganglia [1–5], hippocampus [6–9] and the cortex [10–12] are thought to provide a temporal (ordinal) signal for these action sequences to emerge. However, although seminal modeling work has been carried out to understand how sequences emerge in neural networks [13–15], the mechanistic and dynamic principles by which these neural patterns afford sequential flexibility remain unknown. While several neural network models of corticostriatal circuits exist, these are typically applied to single shot stimulus-action pairings rather than sequential choices, despite extensive evidence that basal ganglia is implicated in such sequential behaviors [16,17] (but see [18] for a nuanced view).

We sought to develop a biologically plausible neural computational model of cortico-basal ganglia circuitry sufficiently powerful to learn arbitrary sequences (e.g., scales) and easily adjust their timing and expression on the fly. In particular, we aimed for the network to be able to learn multiple arbitrary sequences and to allow for temporal asynchrony, shifting, rescaling, and compositionality. We define these terms more precisely below.

Neurocomputational models of sequence production can be broadly categorized in three classes, each with their advantages and disadvantages in computational power and their ability to account for behavioral and neural data.

- In associative chain models (also termed synfire chain [19–21]), activation flows sequentially from one neuron (or neuronal population) to another through feedforward connections [22]. The sequence emerges from the hard-wired structure of the chain. Associative chain models produce sequential and persistent neural activity, as observed empirically [23,24]. They can deal with inherent compression of sequential activity, and learn to produce each action in the sequence at any desired precise time [22]. However, these models are not equipped to facilitate *temporal rescaling*: the finding that learned action sequences can be sped up (compressed) or slowed down (dilated) without the need to overwrite previous learning [25,26]. Moreover, it is unclear how these models implement *temporal shifting*: the ability to start the action sequence earlier or later in time, without modifying the action sequence structure. Chain models also do not straightforwardly allow networks to encode more than a single sequence, given their hard-wired nature.

- Cluster-based models also involve a chained sequence of activation, but this sequence is learned via cell assemblies (clusters) that are chained within a recurrent neural network (RNN), for instance, through spike timing dependent plasticity [27–32]. Hence, the chaining

emerges from synaptic learning rather than being hard-wired. Once the chain is learnt, a single initial input pulse to the RNN induces a sequential activation whereby activation flows from one cluster to another. Cluster-based models allow temporal rescaling [28] while also producing sequential and persistent patterns of activity [27]. Furthermore, they provide a simple mechanism allowing a network to encode multiple sequences. By selectively activating a specific cluster within the RNN, only the cluster "in line" (i.e., connected to the previous cluster) is activated sequentially. Therefore, the RNN can encode multiple sequential actions by learning (and selectively activating) distinct cluster chains encoded in the connectivity matrix [28]. Yet, it is unclear how these models could facilitate action sequences with *temporal asynchrony*: the ability to learn, and flexibly manipulate, motor sequences with varying inter-action intervals (an advantage of associative chain models [22]). Indeed, cluster-based models can flexibly manipulate sequences; however, these sequences are typically iso-synchronous [28]. In addition, although emergent connectivity within and between clusters (or units) can arise via unsupervised learning [33], this connectivity crucially depends on the sequential nature and timing of the teaching input signal to the distinct subsets of the RNN.

- State-space models [34] do not assume a chaining structure at all. Based on a (sparse) randomly connected RNN structure, these models produce a neural trajectory that evolves in high-dimensional space which can be used as a temporal basis to perform a range of complex tasks [35]. However, to reliably reproduce the same task, neural trajectories need to be robust to noise. To that end, state-space models typically harness a noiseless neural trajectory (based on any random connectivity) which is then subsequently used as a continuous teaching signal in presence of noise [34,36,37]. Alternatively, each individual unit in the RNN can be taught via supervised learning to reproduce the neural activity of an empirical dataset [38]. The resultant learned neural trajectory [39] acts as a robust travelling wave that can be decoded by downstream neurons to produce highly complex and flexible motor sequences [40]. However, to reproduce reliable motor sequences, state-space models require highly supervised teaching signals specifying the full neural trajectory and non-biological learning mechanisms (e.g., residual least squares learning algorithms [38,41]). Recent work has shown that biological learning rules using local information can effectively learn complex (sequential) tasks [42–44] (albeit not as effectively as non-biological rules). State-space models can also implement a rudimentary form of temporal rescaling, in that they can rescale the timing of the execution of a single motor response [45], and iso-synchronous action sequences (e.g., index tapping at a steady rhythm) [34]. However, these models do not support temporal rescaling in the more general case (i.e., asynchronous action sequences). Furthermore, given their focus on cortical networks, these models do not address the growing evidence that action sequences unfold over multiple levels within cortico-basal ganglia-thalamic loops, with attractor state switches occurring in the prefrontal cortex [46] and action timing represented in the basal ganglia [2,4].

- Finally, none of the models have tackled how a learned sequence at a particular tempo can be executed with a completely different tempo which may have been learned for a different sequence (e.g., applying a bossa nova rhythm to a rock song). We refer to this ability as *temporal compositionality*.

In sum, all models can account for distinct functionalities in sequence production, but fail to provide a plausible neurocomputational mechanism from which most fundamental abilities–temporal asynchrony, shifting, rescaling, compositionality–can emerge and interact. These limitations arise from a property common to all action sequence models: action identity,

timing and sequence order are represented jointly within the recurrent weights of the network. In the reinforcement learning domain, such joint coding of task features facilitates only rigid forms of generalization and transfer, whereas the ability to code task features compositionally facilitates more robust transfer [47] that can better account for human behavior [48]. However, the mechanisms for such compositionality in neural networks remains unknown.

Here, we develop a biologically inspired (we further discuss biological plausibility of our model in the Discussion) RNN called the associative cluster-dependent chain (ACDC) model. By combining strengths of the associative chain and cluster-based models, ACDC accounts for biological data. As we show below, the novelty in our model is twofold. First, we propose a biologically-plausible model of cortico-basal ganglia-thalamic loops that decomposes the functions of cortex and basal ganglia and learns sequences based on simple local and (biologically motivated) supervised learning rules. Second, this decomposition affords greater flexibility in generating desired action sequences, supporting temporal asynchrony, shifting, rescaling, and compositionality in a single model. Crucially, our model factorizes action sequence features within the circuit, with cortical RNN representing latent states within a sequence, and BG controlling both the timing of the transitions from one state to the next and which actions are linked to sequence positions. Factorizing order and timing information by storing them separately in a premotor cortical RNN, which is dynamically gated by a basal ganglia-thalamus module, affords independent (and flexible) manipulation of sequence order and action timing, and thus increases computational flexibility.

## Results

We start by providing the reader with an intuitive functioning of the ACDC model (Fig 1). The Methods section provides detailed mathematical formulation and further grounds the model within the context of neurophysiological observations on the premotor cortex (PMC) and the BG. The ACDC model comprises a context module encoding the sequences to be executed (e.g., which song is to be played), and is provided as input to a RNN. This input targets a subset of RNN excitatory units, which cluster together via Hebbian learning, encoding the first latent state in the sequence (but not its specific action). In turn, the G (for Go) units in the BG learn (also via Hebbian learning) to link this RNN cluster to the appropriate action (blue arrow 1 in Fig 1), allowing it to accumulate evidence for the first action in the sequence. The G node, part of a G-A-N triplet, projects excitatory connections to its correspondent A (for Action) node (blue arrow 2 in Fig 1) which learns (via a delta rule) weight values for these projections to fine-tune the appropriate timing for this particular action. The A node represents motor thalamus, and its activation has two important consequences. First, it sends a thalamostriatal back-projection to excite the N node (blue arrow 3C in Fig 1), which finally inhibits the G node via lateral connections from D2 to D1 medium spiny neurons [49]. Second, the thalamic A node triggers a transition in the RNN, via a combination of excitatory projections to another RNN cluster (blue arrow 3A in Fig 1), and to a shared inhibitory neuron (blue arrow 3B in Fig 1), consistent with evidence that thalamic units target both cortical excitatory and inhibitory neurons [50–52]. Thus, whenever an action is executed, the ratio of excitatory to inhibitory inputs to the RNN is perturbed in a way that induces a transition from the current cluster to the next cluster in line (targeted by the feedback projections of the current A node, blue arrow 3A in Fig 1) to be expressed (see Methods for more details).

Learning takes place over fast and slow time scales. Hebbian learning is fast and unfolds within the dynamics of a trial (i.e., during the evolution of an action sequence). In contrast, the delta rule is slow and is implemented between trials, via a signed error computed through the discrepancy between the action timing provided by the tutor and the generated action. Action

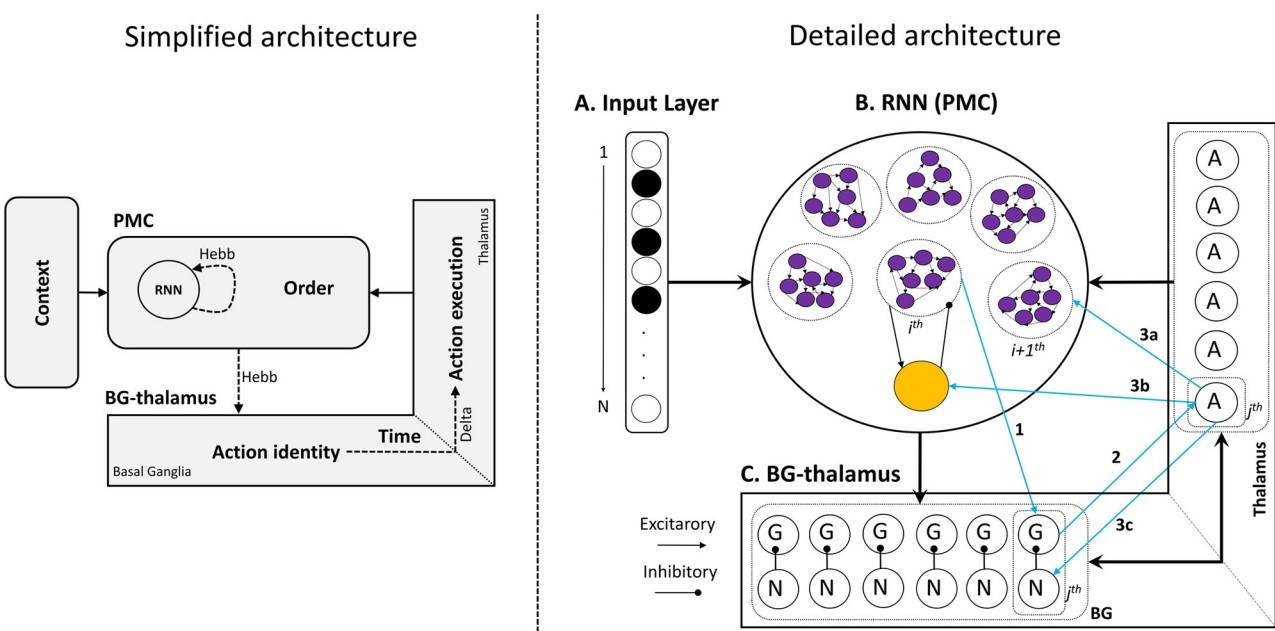

**Fig 1. Simplified ACDC model architecture (left).** An input context layer indicates which sequence needs to be learned or executed. The premotor cortex (PMC) is subtended by a RNN that learns (via Hebbian learning) to form clusters of excitatory neurons encoding order in the sequence, and which are regulated by an inhibitory neuron. In turn, each cluster learns to trigger action plans, topographically represented in the BG. Specific actions are executed in the thalamus at specific times based on learned connections from BG to thalamus. Motor activity is then fed back to the RNN, closing the cortico-basal ganglia loop. The unfolding of several iterations of this loop is responsible for the execution of precisely timed action sequences. Dashed lines are plastic connections, and the associated learning rule is indicated (Hebb = hebbian learning, Delta = delta rule). Note that the plastic connections from action identity to action execution correspond solely to the blue projection n°2 on the detailed architecture figure. **ACDC full model architecture (right). A. Input layer**: codes for contexts indicating the sequence to be learned/produced in a *N* length binary vector. **B. RNN**: represents recurrently interconnected neurons of the PMC, composed of a subset of interconnected neurons (i.e., clusters) that can give rise to sequential activation states after learning via cortico-basal ganglia loops. All excitatory nodes in the RNN project to a shared inhibitory neuron (orange node), which in turn inhibits all excitatory neurons (purple nodes; shown for just one cluster for visual simplicity). **C. The BG**: composed of two neuron types G (Go cells) and N (No Go cells). Go nodes accumulate evidence over time and excite Action (A) nodes in the BG output /thalamus layer. Once activity in a Go node reaches a specific threshold, the corresponding action is executed. Once executed, Action nodes reciprocally activate No Go nodes which in turn suppress Go nodes, shutting down action execution. **The thalamus**: is composed of Action nodes whose activity represents action execution. The $j^{th}$ Action node selectively projects excitatory connections to the $i+1^{th}$ cluster in the RNN, the shared inhibitory neuron and the $j^{th}$ No Go node in the BG. Light blue arrows represent the $i^{th}$ cortico-basal ganglia loop instance. The subindex $i$ refers to the ordinal position in the sequence, i.e. the order the action possesses in the sequence. The subindex $j$ represents the action that is associated to the ordinal position in the sequence.

sequences are learned sequentially: the model learns to produce the first action at the appropriate time, then the second, and so forth. Sequential learning improves motor execution [53–57], and is at the base of several theoretical models of motor sequence learning [58–60].

At a higher-level, order is encoded as a sequence of attractor states represented by persistent activation in distinct excitatory RNN unit clusters (cell assemblies). These clusters do not represent the actions themselves but rather their abstract order; the specific actions to be executed are learned via RNN projections to the BG and their timing is encoded in the weights of topographic projections to the motor thalamus. To optimize precise action timing, the weights between action identity (G unit activity) and execution (timing for a given action conditional on G unit activity) are learned via supervised learning (i.e., delta rule), perhaps summarizing the role of cerebellum in error corrective learning. This allows us to model tasks in which a tutor provides feedback (e.g., [61]; see Methods). Finally, feedback to the RNN from thalamic activity ultimately creates a cortico-basal ganglia loop. Each loop subtends the appropriate action order, identity and timing execution, allowing precisely timed action sequences to unfold. As we show below, our model architecture, allowing to uniquely encode timing

information in a distinct subset of the network (BG) than the one encoding order (PMC), will prove to display advantageous properties. In particular, being able to flexibly control, via external stimulation, the dynamics of the BG will result in a model displaying several temporal flexibility properties.

## Learning precise spatiotemporal sequences

**Simulation 1: Learning temporally asynchronous action sequences.**   Fig 2 shows simulation 1, where the ACDC model learns to produce a precisely timed, temporally asynchronous, action sequence (here, for 6 actions). The goal of the model is to produce each action sequentially at the appropriate time, here at 200, 250, 400, 700, 750 and 900 ms. This is an arbitrarily chosen timing sequence; the model can (learn to) produce any timed, synchronous (see below) or asynchronous, sequence. Fig 2A shows how the activity of each Action node progressively reaches the optimal time (depicted by color coded vertical dashed lines), reflected in a decrease in the action timing error (Fig 2B) and in the weight changes between Go and Action nodes (Fig 2C).

Fig 2D depicts the RNN connectivity matrix after learning (weights are zero before learning). Excitatory projections to the RNN from the input and motor layer are pseudo-random, with the restriction that two different projections never excite the same RNN neuron. These pseudo-random projections make it hard to visually identify the presence of clusters in Fig 2D; importantly however, this connectivity matrix does induce clustered dynamics (see S2 Video). Fig 2E shows how the $i^{th}$ cluster in the RNN learns to be (almost) selectively wired with the $j^{th}$ Go node. Finally, for completeness and transparency, Fig 2F portrays the dynamics of G (left) and N (right) nodes after learning of the sequence.

## Temporal flexibility properties of the ACDC model

Having established learned RNN clusters during sequences, we now focus on the flexibility properties of the ACDC model after learning, without having to overwrite learned weights. First, we show that a previously learnt action sequence with *temporal asynchrony* can be flexibly reproduced. Second, we show that this sequence can be initiated earlier or later in time; we call this property *temporal shifting*. Third, we demonstrate how action sequences can be compressed or dilated, i.e., *temporal rescaling*. Fourth, we show how a given ordered sequence can be produced with a completely different tempo, a property that we refer to as *temporal compositionality*. Fifth, we describe how the model can also output sustained action execution. Finally, we show how the ACDC model can learn (a part of) the Thunderstruck song, which is then flexibly played on a bossa nova tempo; thereby recapitulating the temporal flexibility properties.

**Simulation 2: Reproduction of previously learnt action sequence displaying temporal asynchrony.**   In simulation 1, we demonstrated that the ACDC model can learn precisely timed, temporally asynchronous, action sequences. In simulation 2, we freeze the weights and simply observe that the network can reproduce the sequence maintaining its precision in action timing (Fig 3A).

**Simulation 3: Temporal shifting.**   The previous action sequence can be shifted in time, i.e., initiated earlier or later. Importantly, this shift can occur without changing the timing between actions (i.e., sequence timing is preserved). The ACDC model achieves flexible temporal shifting by implementing an additive positive (to start the sequence earlier) or negative (to start it later) input to the first Go node of the sequence, analogous to the top-down input from pre-SMA to striatum thought to bias starting points for evidence accumulation [62] (although similar effects could be implemented by dopaminergic modulation; see Discussion).

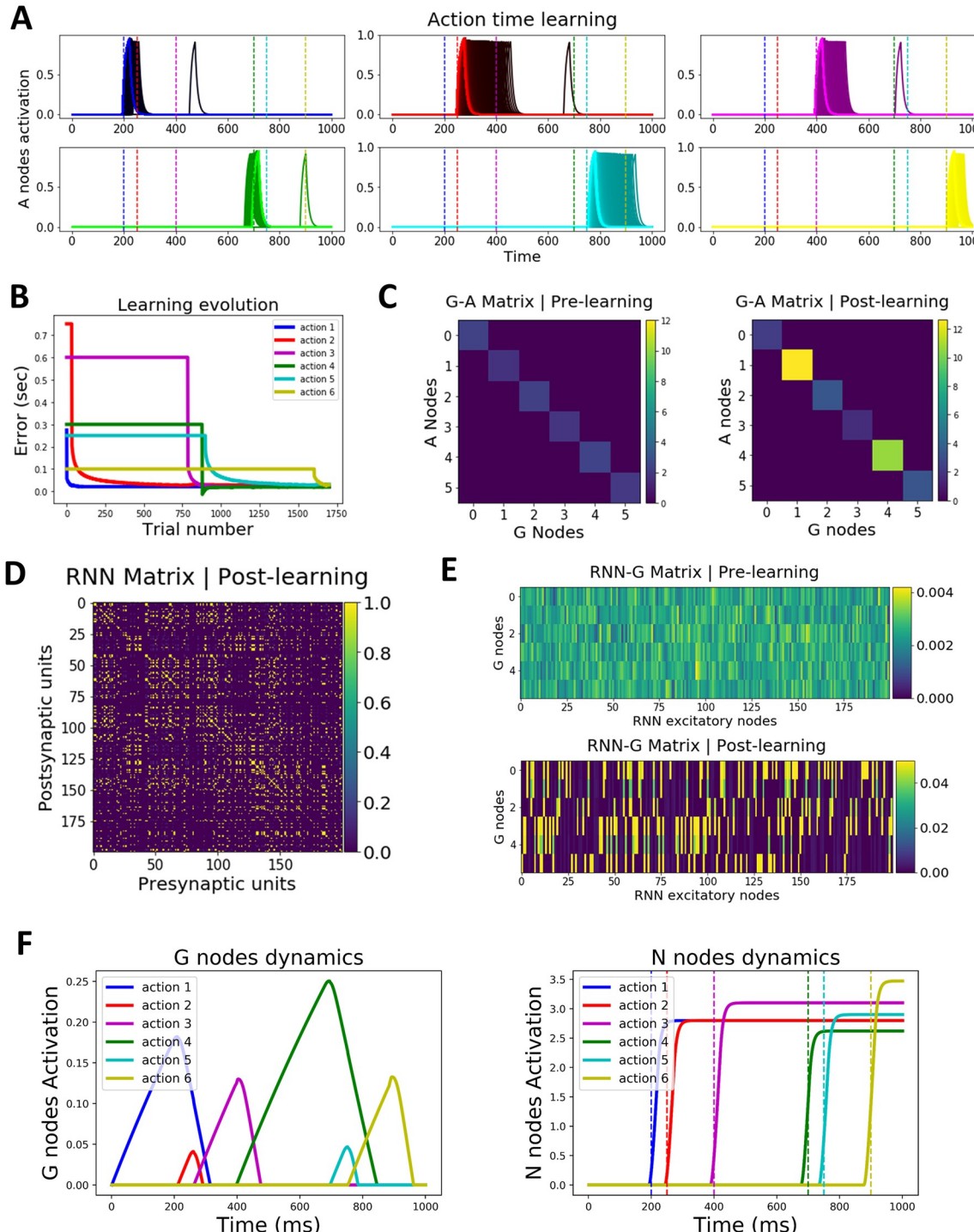

**Fig 2. ACDC's learning dynamics. A. Learning a precisely timed action sequence**. Each action execution (A node activation) is progressively shifted towards the optimal action time (depicted by the color coded vertical dashed line; x-axis represents time). Learning progresses from darker to brightest colors. The 'black trials' are early learning trials where the different shades are not distinguishable **B. Learning evolution**. Color coded traces represent the evolution of the error as a function of trial number for each action in the sequence. Learning unfolds sequentially, whereby timing errors are minimized for the first action before the second action starts learning. Therefore, each action (except action 1) starts off with a plateaued error level until the preceding action reaches the optimal time. Some action timings are learned faster than others because their optimal time weight value is closer to their initial value. The error is computed by subtracting the observed from the desired response time and plotted in seconds. **C. BG weights encode time**. Action timing is learned by changing the weights from BG Go nodes to thalamus Action nodes. The left and right panel show respectively the weights

values before and after learning. For instance, the second action (red trace in B) starts off being produced too slowly. Hence, weights increase until they produce the optimal action time for action 2. Color bars indicate weight values. **D. RNN connectivity matrix after learning**. The RNN connectivity matrix is initialized as a blank slate (all values are set to 0). After learning, the RNN connectivity matrix displays the appearance of clusters, whereby groups of 20 neurons are fully interconnected with each other and not connected with other neurons in the RNN (please refer to S2 Video for better visualization of clusters and their transitions as the sequence unfolds). Color bar represents weight values. **E. RNN $i^{th}$ cluster learns to project to $j^{th}$ Go node (see Fig 1, 'Detailed architecture', for the meaning of the indices)**. The top panel shows the randomly initialized weight values between the RNN excitatory units (before learning). The bottom panel shows how each cluster (represented by a subset of RNN neurons) is connected to a specific Go node after learning. Color bars represent weight values. **F**. Dynamics of G (left panel) and N (right panel) nodes after learning.

In simulation 3, we inject an additional input of +1 or -1 to the first Go node during the first 100 ms of the 1 second time window. Fig 3B shows how the sequence is shifted earlier in time for the positive input (left panel) and later in time for the negative input (right panel). Moreover, Fig 3C shows that as this additional input lasts longer, the distance (in time) between the first action of the shifted sequence and that of the original sequence increases linearly.

**Simulation 4: Temporal rescaling.**    Musicians can learn a rhythm, i.e., a precisely timed action sequence, and instantly temporally rescale (compress or dilate) that rhythm without additional learning. In our model, flexible rescaling is achieved by sending a multiplicative input ($\rho$) to all Go nodes simultaneously (i.e. the $\rho$ parameter multiplies the net input in Eq 5, see Methods); if $\rho > 1$ or $0 < \rho < 1$ the sequence is respectively compressed or dilated. Fig 3D shows temporal rescaling for $\rho$ values of 1.2 (compression, left panel in Fig 3D) and 0.9 (dilation, right panel in Fig 3D). Importantly, temporally rescaling the sequence does not affect the temporal structure of action sequences. For 100 values of $\rho$, ranging from 0.9 to 1.2, we computed the relative ratio between a sequence of 3 actions. The ratio was computed by subtracting the time of action 1 from that of action 2 (subtraction 1), then the time of action 2 from that of action 3 (subtraction 2), and dividing subtraction 2 / subtraction 1. We performed this computation for the action triplets 1-2-3, 2-3-4, 3-4-5 and 4-5-6, and summed the ratios. Fig 3E shows that this sum of ratios stays constant (mean = 7.5, s.d. = 0.12), thereby indicating that temporal structure is maintained albeit rescaled. Note that rescaling also induces a tiny shift in the sequence. This is an emergent property of a global rescaling signal to all Go nodes of the network; this slight shift could be avoided by targeting all but the first Go node with the multiplicative term.

**Simulation 5: Temporal compositionality.**    Musicians must also be capable of temporal compositionality; that is, applying a desired tempo to an action sequence that was learned in a different tempo (e.g., apply a bossa nova tempo to a rock song; see below). In simulation 5, we assume that desired tempos are learned and extracted from other sequences, which then can then be used as a dynamical multiplicative input signal to all Go nodes (Fig 3F right panel). Therefore, whereas temporal rescaling makes use of a constant $\rho$ multiplicative input value to the Go nodes, temporal compositionality is achieved by a dynamic multiplicative input (the $\rho$ multiplicative input value follows the blue trace in the right panel of Fig 3F). The result is to produce the learned sequence (described in Fig 3A) to the tempo described by the multiplicative signal. Fig 3F (left panel) shows how the time of each action in the sequence does not accord with the learned tempo (color coded vertical dashed lines), but is rather produced at the novel desired timing (vertical solid lines).

**Simulation 6: Sustained motor activation.**    The ACDC model is also capable of producing sustained motor activation for any element within the sequence, for instance sustained notes in a musical scale. Our model can achieve sustained motor activation via two mechanisms. First, via a flexible mechanism similar to that of rhythm compositionality, a multiplicative signal ($\rho = 0.1$) is sent to the Go node during the period in which sustained motor

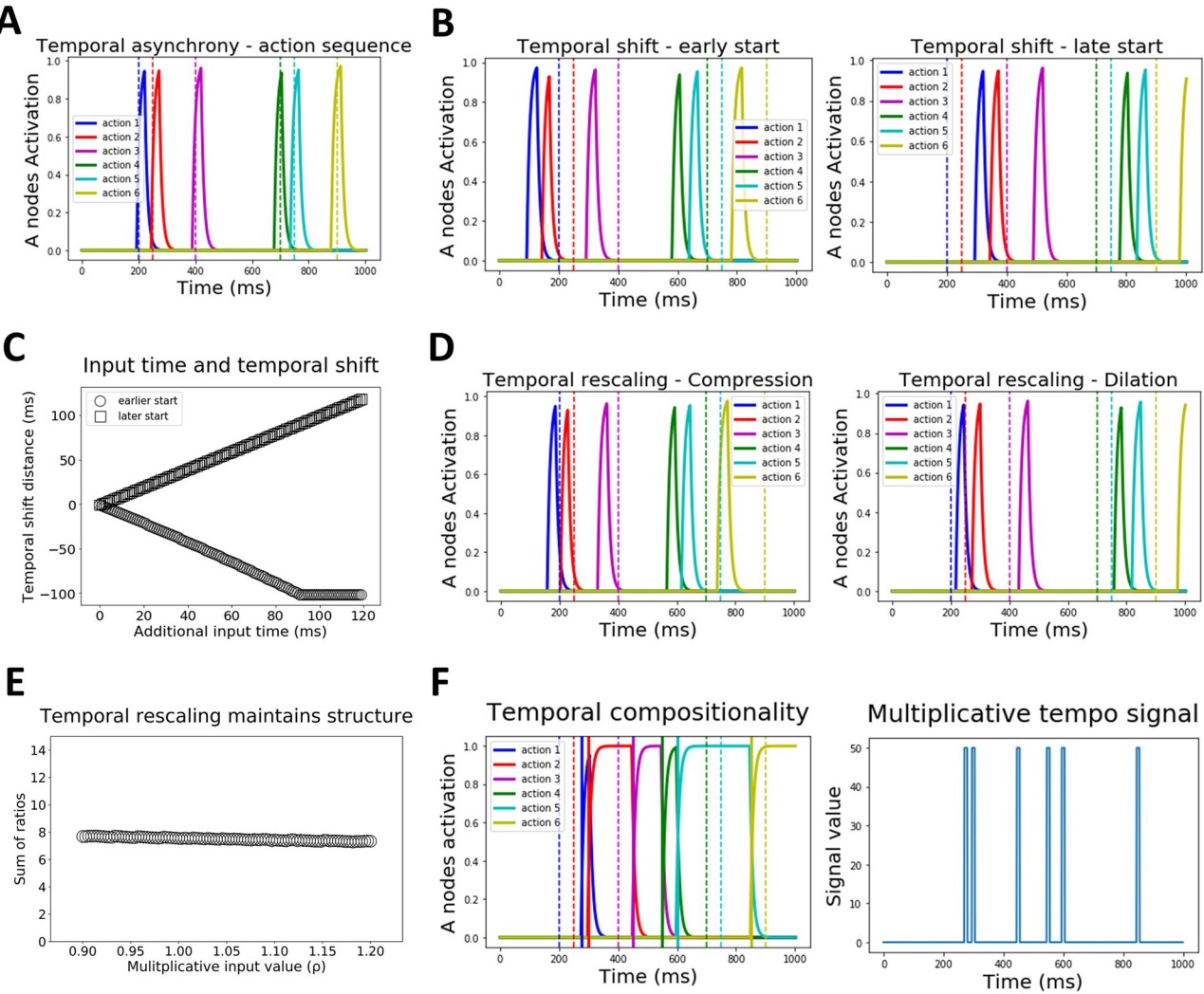

**Fig 3. Temporal properties of the ACDC model. A. Simulation 2: Reproduction of action sequence with temporal asynchrony**. Each action (i.e. A node activation, color coded) is produced at the precise desired time indicated by the vertical dashed line (also color coded), within a 1 second time window. Inter-action interval varies as the sequence unfolds. **B. Simulation 3: Temporal shifting**. A precisely timed action sequence can be started earlier (left panel) or later (right panel) by respectively injecting an additional positive or negative input to the first G node (i.e. associated to accumulating evidence in favor of the first action). Importantly, the temporal structure of the action sequence is not altered. **C. Simulation 3: Temporal shifting varies linearly with additional input time**. Applying longer input times leads to increasingly earlier or later shifts in sequence initiation times, depending on whether additional input is positive (circles) or negative (squares). **D. Simulation 4: Temporal rescaling**. Action sequences can be compressed (left panel) or dilated (right panel) by adding a multiplicative input to all G nodes simultaneously. **E. Simulation 4: Temporal rescaling preserves action sequence structure**. Importantly, when temporal rescaling is applied to the action sequence, the relative timing between each action (i.e. the structure) is preserved. Here, we plot the sum of ratios (y-axis, see main text) as a function of the multiplicative input $\rho$ (x-axis). The sum of ratios value (black circles) stays constant as a function of $\rho$, indicating a preserved temporal structure even though the sequence is rescaled. **F. Simulation 5: Temporal compositionality**. The left panel shows how A nodes activity are activated on the tempo described by the multiplicative signal (left panel). Vertical dashed and solid lines on the left panel indicate the timing of each action for the previous and novel tempo respectively. As shown, the respective A nodes become active on the novel tempo.

activation is needed. Fig 4 left shows that applying such a signal between the second and third actions allows motor activation of the second Action node (red trace) to be sustained until the third action is executed (purple trace). Second, via a learning mechanism, the weight value between a specific Action-NoGo node pair can be decreased to induce sustained activation of the Action node (Fig 4, right). Note that we focus here on the mechanistic properties of the

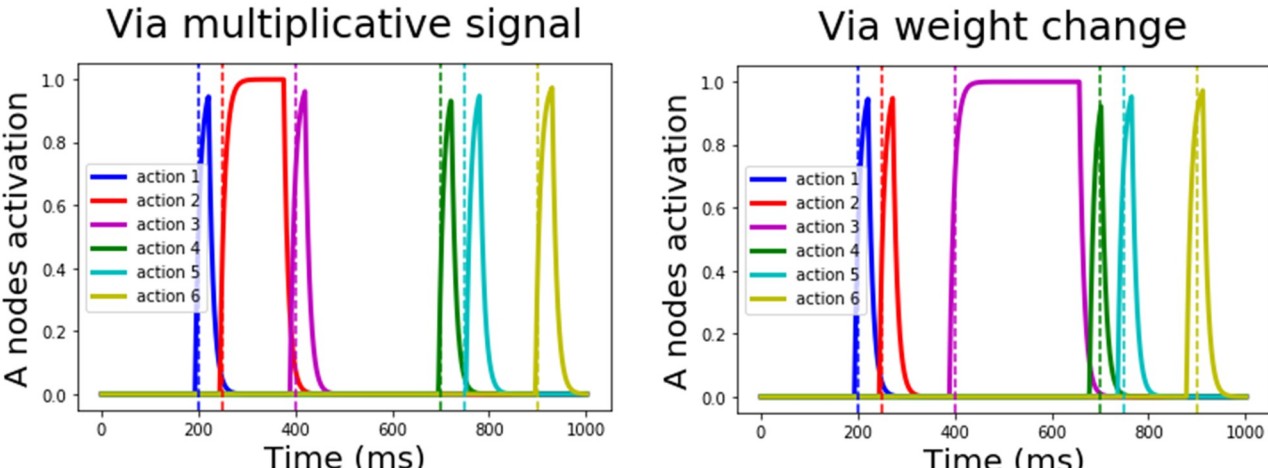

**Fig 4. Simulation 6: Sustained motor activation.** Both panels demonstrate that the ACDC model is able to output sustained motor activation as desired within a sequence. The left panel shows the results of applying a multiplicative signal ($\rho = 0.1$) to the second No Go node, inducing a sustained activation of the second action (red trace). The right panel shows a similar effect this time by decreasing the value of the Action-No Go connection of the third action, in turn inducing sustained activation of the third Action node (purple trace).

model, rather than proposing how the Action-NoGo weights may be learned to support sustained activation.

**Simulation 7: The ACDC model in action and sound.** Here, the ACDC model learns to produce the second guitar riff of ACDC's (the rock group) Thunderstruck song. This riff is composed of 16 actions hitting six different notes (B5, A5, G#5, F#5, E5, D#5) following an isosynchronous rock tempo (Fig 5A). By allowing the model to record each note corresponding to each sequential action (following Fig 5A), the ACDC model was able to musically reproduce the riff (S1 Audio file). Notably, S1 Video shows that the RNN dynamics represent sequential attractor states, encoding order and leading to the production of each action (and sound) in the sequence (for a slowed down demonstration of similar dynamics with a less complex action sequence see S2 Video below). Next, we leveraged temporal compositionality to allow the ACDC model to play the riff but now based on a bossa nova tempo without further training

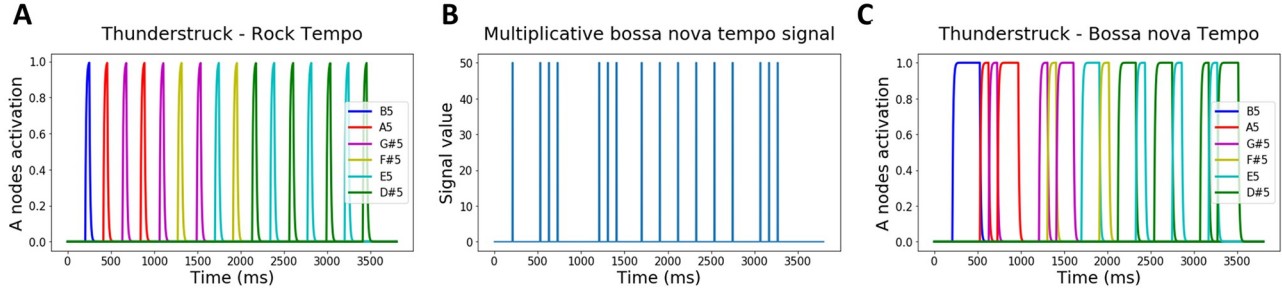

**Fig 5. Simulation 7: the ACDC produces the Thunderstruck song. A. Second guitar riff from ACDC's (the group) Thunderstruck song**. The riff is composed of 16 sequential actions creating a isosynchronous rock rhythm over a window of 3500 ms (given a 140 bpm tempo). Each action is associated to a color coded note). **B. Generic bossa nova tempo**. We imposed the model to replay the thunderstruck rock tempo song following a bossa nova rhythm whose tempo is described by the blue trace multiplicative signal. **C. Flexible generation of the Thunderstruck song following a bossa Nova tempo**. When the multiplicative input (Fig 3B) is given to the Go nodes of the BG, the ACDC model flexibly reproduces the Thunderstruck song but now following the bossa nova tempo.

(Fig 5B and 5C and S2 Audio file). Further note that, altogether, this simulation encapsulates distinct temporal flexibility properties. First, flexibly reproducing the Thunderstruck song following a bossa nova tempo requires the ability to generate an action sequence with temporal asynchrony. Second, temporal rescaling is applied to parts of the song as the sequential execution of consecutive notes need to be sped up or slowed down. Third, the model displays its ability to produce sustained motor activation (see S2 Audio file).

### Behavioral and neurophysiological simulations

**Simulation 8: Behavioral simulation.** In the motor timing literature, a ubiquitous finding is *scalar variability*: when asked to produce an action after a specific time interval, the variability in action execution timing increases with the length of interval timing [63–66]. In simulation 8, our model learns to produce a single action at distinct interval timings (i.e. 200, 400, 600 and 800 ms). For each timing, the model produces 500 reaction times (RTs), from which we extract the standard deviation (SD), and reproduce this process for 100 simulations and two noise values (gaussian random noise with zero mean and SD of 0.01 or 0.05 is added to the model Eqs 3–5 and 7). Reproducing empirical patterns, Fig 6 shows that the SD of RTs increases as a function of interval timing, and thereby demonstrates that the ACDC model displays scalar variability (see also [67]). Furthermore, the SD value range also increases with noise values. This effect is explained in our model by having a fixed negative bias on the Action nodes in the motor layer. Such a feature reduces to having an accumulation-to-bound process for action execution. Hence, given a specific amount of noise, longer RTs are associated to wider RT distributions (i.e. larger SD, [68]). The underlying reason is that the effect of noise on evidence accumulation is amplified as time elapses.

**Simulation 9: Neurophysiological simulations.** Two other ubiquitous findings are persistent and sequential neural activity. First, several studies have observed persistent neuronal firing rates in temporal [69–71], parietal [72–75], premotor [76] and prefrontal [77–79] cortices whenever an agent has to hold in working memory task-relevant stimulus features (e.g., spatial location). Theoretical work suggests that persistent activation patterns emerge from recurrently connected networks that settle in one of multiple potential attractor states [80–82]. Second, as motivated in the introduction, sequential activity has also been observed in distinct sequential behaviors such as spatial navigation [8] and bird song [83–86].

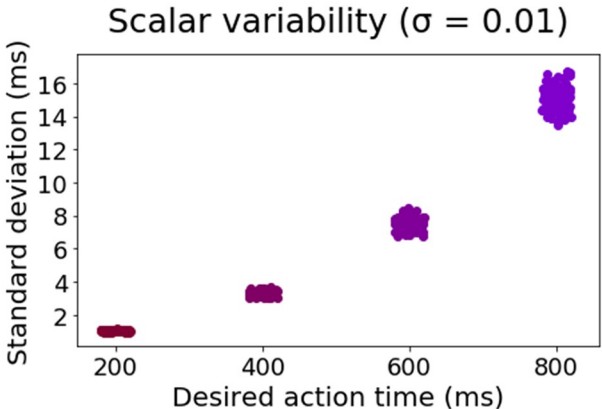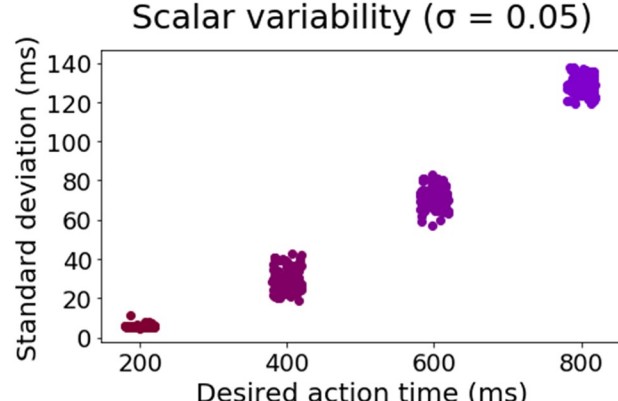

**Fig 6. Simulation 8: The ACDC model displays scalar variability.** Left (low noise value = 0.01) and right (high noise value = 0.05) panels show that the standard deviation of RTs increases as a function of the desired action time (i.e. interval timing). Moreover, higher noise values increase the range of standard deviation. Each dot is the result of 1 out of 100 simulations for each interval timing.

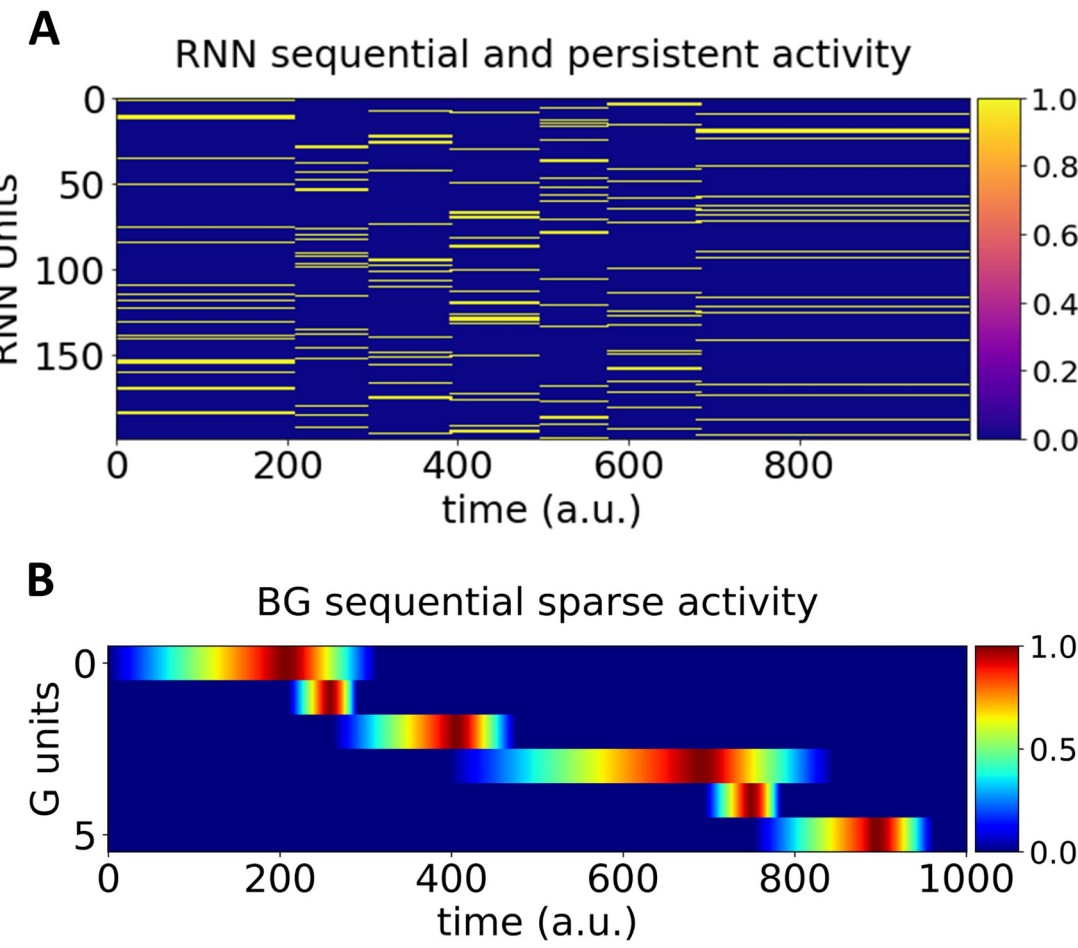

**Fig 7. Simulation 9: A. Sequential and persistent activation of clustered neural populations within the RNN**. The y-axis represents each RNN unit, the x-axis represents time. The first cluster is activated by the input layer, and maintains persistent activity until the first action is executed. At that moment, via excitatory projections from the Action nodes (Fig 1C) to the following ($i+1^{th}$) cluster in the RNN (Fig 1B) gets activated, and thus displays persistent activation, and so forth via the cortico-basal ganglia loops (light blue arrows in Fig 1). Color bar represents firing rate. **B. Sequential and sparse activation in the BG**. The y-axis represents the G unit activity over time (x-axis). Each G unit responds in a sequential and transient manner, as has been shown in neurophysiological single-cell recordings of the BG (e.g., [4]). Color bar represents normalized firing rate.

Interestingly, recent work suggests that sequential switches in attractor states (and hence persistent neural activity) occur during behavioral switches in action sequences [87]. Therefore, persistent and sequential activity may emerge from the same mechanism. In our model, the RNN activation dynamics display such switches from one attractor to another as the action sequence unfolds. Each attractor state is associated to the persistent activity of neurons forming a cluster in the PMC (RNN). When the action associated to that attractor state (i.e. the $j^{th}$ action associated to the $i^{th}$ order) is executed, this triggers a switch in attractor state in the RNN (via cortico-cortical projections from M1 to PMC), as empirically observed [87]. In simulation 1, the ACDC learns to produce an arbitrary sequence of 6 actions, each with their own desired execution time within a window of 1 sec (i.e. at 200, 250, 400, 700, 750, 900 ms). Fig 7A shows the RNN dynamics after learning. Each cluster of activation displays persistent neural activity until the action is executed, which triggers the following cluster of persistent neural activity. Hence, activity in the RNN is both persistent and sequential in nature.

To gain better visual intuition on the RNN dynamics, we performed dimension reduction on the row space of the unit (i.e., neuron) by time matrix displayed in Fig 7A. We then dynamically plotted the first 3 principal components (PCs) as a function of time. S2 Video shows that each cluster of persistent neural activity acts as an attractor state (within the highly dimensional space of the RNN), and the dynamics in the RNN switch from one attractor to the other when an action is executed, again displaying both persistent and sequential neural dynamics.

The qualitative pattern of the RNN sequential and persistent dynamics (Fig 7A) is different than observed in rodent [2,8,11] or monkey [1] neurophysiological recordings, which reveal sequential sparse activation (individual neurons display quick and transient activation as behavior unfolds). Notably however, the Go nodes in the BG module of our model display qualitatively similar sequential and sparse activation patterns as that seen empirically in the BG (Fig 7B; see Figs 2A, 3B, 1E and 8C, respectively of [2,3,5,88]).

## Model regimes and robustness

A useful model should be robust to variations in its key parameters and/or should exhibit regimes in which the model exhibits qualitatively different features [23,89–91]. To examine the sensitivity of our main results to parameter variations, we start by exploring the model regimes in a network with fixed connectivity. We focus on the recurrent weights within the excitatory RNN units, as well as the weights from the G to A units, which we consider similar to recurrent and feed-forward weights in traditional associative chain models [24]. We explore the necessary weight combinations for the model to execute an entire sequence of 6 actions, within a temporal window of 1 sec (as described in the learning subsection). Fig 8A shows that weight values from G to A units subtend the number of actions that can be executed within the 1 sec temporal window: higher G to A weights progressively lead to sequences with more actions. Notably, recurrent weight values do not influence the model's ability to produce the sequence (no impact of variation along the x-dimension). In contrast to previous models [23], recurrent and G to A weights (i.e., feed-forward weights) do not control the model regimes, i.e., whether the model produces damping, sequential or persistent activity. This difference emerges from divergent architecture between our and previous models. Persistent activation of each action during the entire sequence cannot emerge in our model. Indeed, action execution is automatically associated to a switch in attractor state within the RNN. This hinders the previous state to continuously activate its G unit and hence cannot provide evidence for the previous A unit. Therefore, if any, persistent activation of an action can be maintained only up until the next action is executed.

As shown in simulation 6, one way of controlling persistent activity is through the weight values of the A to N unit projections: activation of a particular action shuts down that action via feedback projections to N units, thereby reducing persistent activation. To quantify persistent activation across an action sequence, we derive a "sustainability" measure by computing the area under the curve (AUC) for each A unit activity that falls above the value of 0.5, summing the individual AUCs, and normalizing by the total sequence time (i.e. 1000 ms). Fig 8B shows a heatmap of this measure as a function of A-N and G-A weight values. Sustainability increases with lower values of A-N weight values (note that we plot this measure only for parameter combinations that produced a full six action sequence; otherwise we set this value to 0). Interestingly, the entire range of sustainability is covered by the A-N weight values over a broad range of G-A weight values, from transient pulse (blue values) to sustained actions (red values).

As noted above, a prominent feature of our model is that action execution triggers switches in RNN attractor states through parallel projections to its excitatory and inhibitory units. For

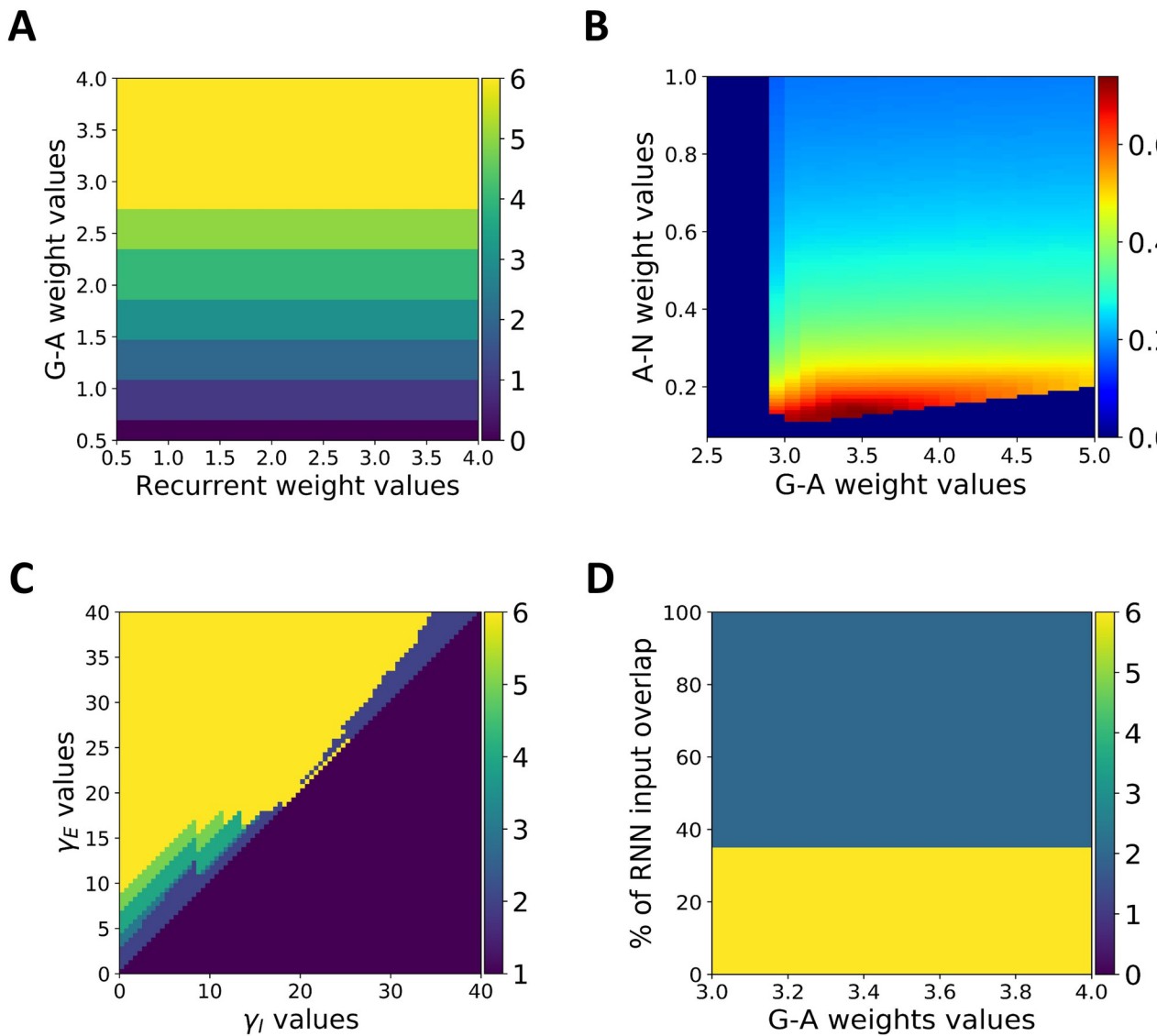

**Fig 8. Model regimes and robustness. A. Number of actions within a sequence as a function of feedforward and recurrent weights**. The y- and x-axis represent G-A and recurrent weight values, respectively. As depicted, the G-A weight values control whether the model can produce six actions within a 1 sec temporal window, irrespective of the recurrent weight values. The color bar codes the number of actions that are produced within the sequence; yellow for a full sequence (six actions) and dark blue for no actions. **B. Action sustainability**. The heatmap reflects the sustainability measure magnitude (warm colors coding for higher values) as a function of A-N and G-A weight values. As depicted, action sustainability increases with decreasing values of A-N weights, over a large range of G-A weight values. **C. Model regime as a function of the $\gamma_E$ and $\gamma_I$ parameters**. The y- and x-axis represent $\gamma_E$ and $\gamma_I$ parameter values, respectively. As depicted, the model can reproduce fully and precisely a learnt action sequence within a broad range of parameter combination respecting the $\gamma_E > \gamma_I$ inequality. Color bar is identical to A. **D. RNN input overlap**. The model can produce a full six action sequence up to 35% of input overlap to the RNN between contiguous actions; i.e. activating 35% of the previous and subsequent cluster in the RNN. Stronger overlap leads to a break in the sequence after 2 actions. Color bar is identical to A and C.

the switch to take place, the gain must be stronger on projections to RNN excitatory vs inhibitory units (i.e. $\gamma_E > \gamma_I$), allowing the previous cluster to shutdown while also activating the next cluster in line. To explore the robustness of this property, we used the learned sequence in simulation 1, and parametrically modulated both gain values ($\gamma_E$ and $\gamma_I$ values are represented on the y and x dimension, respectively). As depicted in Fig 8C, a vast range of gain

combinations allows the model to fully and precisely reproduce the learnt sequence (yellow space), as long as $\gamma_E > \gamma_I$. Note that some combinations allow for a partial reproduction of the sequence.

Finally, one simplifying assumption in the model is that feedback inputs to the RNN are orthogonal and do not overlap (i.e., projections from distinct A units never activate the same RNN excitatory units). In Fig 8D we relaxed this assumption and progressively increased the percentage of overlap between contiguous A unit inputs to the RNN. We observed that the model can continue to produce a complete six action sequence as long as the overlap between inputs of contiguous actions does not surpass 35%, at which point the model fails to reproduce a full sequence and only manages to produce the two first actions as two quick pulses. Further, note that the effect of input overlap is independent of the G-A weight values (taken from a range that produces a full sequence, see yellow area in Fig 8A).

## Discussion

The ACDC model combines the strength of associative chains (e.g., [23]) and cluster-dependent (e.g., [27]) models, while also formalizing how the BG contribute to recurrent cortical dynamics in sequential behaviors. Our model factorizes action order, identity, and time, which are represented in distinct loci of the cortico-basal ganglia neural network. Crucially, factorizing these features provides the network with the ability to independently manipulate the building blocks of precisely timed action sequences, thereby increasing the computational power of our model. This increased power is illustrated through several interesting emergent properties. First, we demonstrated that the ACDC model can learn and reproduce precise spatiotemporal action sequences with temporal synchrony or asynchrony. Second, our model displays several flexibility properties: temporal shifting, rescaling and compositionality, and sustained motor activation; culminating in our model's ability to reproduce the Thunderstruck song and change it to a bossa nova tempo. Third, the model can account for behavioral and neurophysiological empirical observations. Finally, we showed that the main model properties are robust across a range of parameter values.

### Encoding order as attractor state switches in the RNN

Recent work suggests that dynamic representations can be understood as switches in activity of neural networks [92], whereby action sequences emerge from neural attractor states unfolding over time. Indeed, Recanatesi et al. [46] showed that sequential behavior was subtended by the sequential unfolding of attractor states in rodent secondary motor cortex. Furthermore, these authors modeled variability in action timing by adding correlated noise to the dynamics of a RNN, leading to dynamics that jump from one attractor state to another at random times (hence explaining the variability in action timing). In our model, the switch between cortical attractor states is not random but is controlled by dynamic BG modulation of excitatory thalamic projections to the RNN that transiently modify the ratio of excitatory to inhibitory inputs. Within our conceptualization, we suggest that persistent activity within a cluster indicates the latent state that the system is in [90], which in this case reflects the ordinal position in the sequence. Moreover, in contrast to previous models of frontal cortical BG interactions [90], the cortical clusters themselves were not assumed to be anatomically hard-wired but emerged within the RNN via learning.

Alternative models have proposed different mechanisms for encoding ordinal position. Some models possess a temporal context layer whose state is modified dynamically as time passes [93–96]. Other models assume that the network input (used to learn the sequence) is itself sequential in nature [27,28], and learning the spatiotemporal signal depends on the

sequential nature of the input. Our model is free of this assumption; the network input is a single pulse of activation, but can nevertheless reproduce a precisely timed spatiotemporal signal. This ability emerges from the feedback loop from thalamic Action nodes to the cortical RNN, triggering transitions to a subsequent cortical attractor. One can therefore consider motor output as part of the teaching input signal to the RNN; because motor activation unfolds sequentially in our model, the sequential nature of the teaching signal emerges from our network architecture.

Interestingly, the idea that the motor cortex (presumably via motor thalamus neurons) acts as a teaching signal to other brain areas has received strong support from rodent lesion studies. For instance, rats are unable to learn a precisely-timed lever press when their M1 cortex is lesioned [97], and transiently inactivated or disturbed via optogenetic manipulation [98]. More generally, the notion that motor output can influence cognitive representations and transitions is consistent with the emerging literature on how cognitive functions scaffold on top of motor functions in cortico-basal ganglia circuits [99,100].

## Motor sequence flexibility as inputs to the basal ganglia

Humans can adapt their motor output almost instantaneously given external or internal stimuli. For instance, musicians can modify the tempo of a song upon signaling of the conductor. Such flexibility necessarily needs to stem from fast reconfiguration of neural dynamics, rather than emerge from changes in networks weights [12]. Murray and Escola [28] proposed a model of interconnected medium spiny neurons in the striatum that can apply such dynamic reconfiguration. In particular, their model could perform temporal rescaling of sparse sequential activity. Yet, flexibility in this model is constrained to isosynchronous sequences (see also [67,101]). However, a recent model [22] making use of eligibility traces [102–106], manages to learn precise asynchronous spatiotemporal sequence learning. Still, it is unclear how such a model can rescale asynchronous sequences, and neither of these models is capable of exhibiting temporal compositionality. Nevertheless, ring-like models with synaptic depression [e.g., 28] could potentially account for these properties. Indeed, temporal rescaling in these models is often implemented as changes in the background current (higher current levels lead to faster rescaling). Therefore, to produce asynchronous sequences, one could imagine a dynamical background current which is null when no action has to be provided, resulting in a silent network, and turned on when the sequence has to be resumed. However, given a silent noisy network, reactivation of a global current will induce activation of a random cluster in the sequence and not necessarily the next cluster in line, thereby not displaying robustness in sequence production. Indeed, for such a system to be viable, the network would need an additional piece of information, which is a memory of which cluster (or unit) was last active. The reason is that, in ring-like models the background current is global (i.e., sent to all units). Thus, if the network goes silent, reactivation of the network would not ensure that the appropriate cluster (i.e. the one next in line) becomes active; i.e. noise would randomly activate a cluster and the sequence would restart from that cluster (see implementation of [28]). Consequently, some extra input signal (representing a memory of which cluster was last active) should give an advantage to the next cluster in line, in order to ensure that sequence order is maintained. This extra input signal could take the form of recurrent connections within the cluster, controlling the activation decay of that cluster such that it would still be active after long periods of time (as opposed to all other clusters that would be fully silent; see [22]). Note however that recurrent connections would put a break on the synaptic depression, as these two parameters trade off; therefore, formal implementation of this proposal would be needed to understand the model regimes.

Crucially, the ACDC model can perform temporal rescaling for both iso-synchronous and asynchronous sequencing, and it can also flexibly switch the tempo altogether through a multiplicative signal to the BG. Our model proposes a more robust approach to sequence flexibility. The sequence in our model is chained by the execution of the previous action, thus our model does not require an explicit representation of the most recent action. Indeed, what controls the sequence is a series of local signals (i.e., selective feedback from the motor thalamus to the RNN [49]) rather than a global signal, and what controls the timing is a global input to the BG. This feature is key for our model to account for the distinct types of flexibility described in the results.

It is important to clarify that flexibility in our model is implemented at the action selection level rather than action execution or implementation. Indeed, although we use musical examples to motivate our work, our model focuses on the timing/selection of actions rather than their execution. Even simple finger movements implemented to play the piano require muscle commands represented as highly dimensional and continuous signals [40,107,108].

The temporal properties of our model discussed above emerge from additional inputs to the BG. What is the nature of this input? One possibility could be dopaminergic. Indeed, midbrain dopaminergic nuclei massively broadcast to the striatum [109], and several studies have implicated dopamine in controlling movement vigor [110–118]. Dopamine has also been extensively implicated in impulsive (i.e. pathologically speeded) behavior [119–125]. Furthermore, administration of amphetamine and haloperidol to human participants, respectively increasing and decreasing tonic dopamine levels, has been associated to faster and slower response times during a simple reaction time task [126].

If dopamine can flexibly modulate (speed up or slow down) action execution timing, the question remains upon which psychological process this neuromodulatory effect takes place. Within the accumulation-to-bound framework [68,127], this effect could potentially alter two distinct processes. First, dopamine could play a role on the speed (or rate) of evidence accumulation. In line with this hypothesis, several studies have highlighted a clear effect of dopamine on the drift rate of evidence accumulation in perceptual [128,129] or reward-based [130] decision-making tasks. Our model implements this possibility. Indeed, inputs to Go nodes modify (increase or decrease) the drift rate of evidence accumulation. Yet, the speed at which an action is produced also depends on the response threshold, with lower thresholds increasing speed at the expense of accuracy [131]. Therefore, a second alternative is that dopamine or other BG modulations may modify the threshold of action execution [132,133]. Interestingly, Parkinson's disease patients on subthalamic deep brain stimulation tend to behave impulsively [125], due to modulation of the decision threshold [134–136]. Naturally, both hypotheses are not mutually exclusive; further research should investigate the effects of dopaminergic and subthalamic modulations regarding motor sequence flexibility.

## Limitations and future directions

As previously noted, some of the implementational and biological details of our model remain to be worked out. First, we simplified the BG gating circuitry, to focus on the G and N populations, summarizing their effects on downstream thalamus but omitting the disinhibitory circuitry involving the substantia nigra. Many previous models, including our own, have simulated the more complete direct and indirect pathways but we did not feel this detail was necessary for the present purposes. Second, reinforcement learning of action timing is conceptually thought to take the form of a three-factor hebbian learning rule [43,103,137–139], where neurons subtending a rewarding behavior (and hence forming a specific cortical activity patterns) increase their connectivity to D1-receptor containing striatal "Go" populations via

dopaminergic activity bursts stemming from midbrain nuclei [90,140–143]. While we do not challenge this mechanism, we focused learning in our model to one synapse downstream, from Go nodes to thalamic motor neurons. Indeed, recent evidence suggest that error-driven learning can be achieved via manipulation of BG outputs to thalamus [144]. Third, although much evidence indicates that the BG learn via reinforcement learning [145–147] (i.e., depending on whether rewards are better or worse than expected), we incorporated a signed error in our learning mechanism which is more powerful for timing signals. The supervised delta rule is only used at the output of our network to optimize readout weights. Thus, we do not view this learning rule as implausible in our model given that we focus on sequence learning situations in which signed error feedback is provided (see [61]), such as when a tutor teaches you how to play the drums and holds the tempo, or in bird-song learning [84] (where a tutor is available to provide signed error). There are several biologically plausible implementations of the delta rule when such error signals are available (e.g., [148,149]). Our learning rule in the BG-thalamus thus summarizes the contributions of these systems in conjunction. Nonetheless, future work should investigate how and whether complex precise sequences may emerge solely based on reinforcement learning. In principle, the ACDC model could learn sequences solely based on RL. However, learning would be much more tedious [43]. In contrast to ACDC, other models have directly trained the entire time-series of individual RNN units to match empirical data [38,150], using continuous learning signals (also see [34,36,37,39]). Hence, the neurophysiological simulations in ACDC (i.e., sequential sparse activation and attractor states) emerge only from the proposed theoretical architecture in the context of behavioral experience. Fourth, we implemented a single inhibitory neuron in the RNN-PMC module. Our focus was on the functional role of inhibitory neurons on the transition of attractor states [28], and certainly this single unit could be replaced by a larger population. Future versions of the model should include a broader pool of inhibitory neurons in the RNN as they have been shown to exhibit mixed selectivity to multiple aspects of a task [151]. Fifth, one assumption in our model is orthogonality of all projections to the RNN. Although we showed that this assumption can be relaxed to a certain extent, this feature ensures there is no ordinal interference during action sequence execution. In the brain, this orthogonality may be implemented via mixed selectivity of excitatory frontal neurons that ensure downstream readouts without interference [152,153]. Interestingly, Márton et al. [154] recently developed a RNN model of cortico-striatal interactions optimized to learn oculomotor sequences. Similar sequences were performed by awake monkeys while activity was recorded in their dorsolateral prefrontal and striatal areas. Learning to implement the correct actions for each sequence pulled apart the representational structure of action sequences in activity space both in the model and neuronal recordings. Whereas ordinal representations in our network were hardwired as orthogonal vectors in the RNN (in order to avoid interference), the work of Márton et al. suggests this may emerge naturally through learning.

Our model simulates action sequences such as those needed to play the guitar or the piano. Within this context, each action is represented as a discrete entity. However, many daily life action sequences are subtended by more continuous actions, as for instance when playing violin with a bow. The ACDC could be expanded by having more continuous representations of action plans and execution in our BG-thalamus module. Based on dynamic field theory, one potential approach would be to represent actions as dynamic neural fields [155–157], which have been shown to successfully model more continuous reaching actions [158]. Moreover, these continuous action representations in the BG may require additional inputs from the cerebellum for movement coordination [159] or sequence prediction for motor control [160]. Moreover, our model (as others [22,28,67]) was specifically engineered to account for spatio-temporal sequences and how these may be flexibly manipulated. This in contrast to other

instantiations [35,161] of RNNs (i.e. reservoir computing) that find natural solutions to diverse tasks involving distinct psychological processes (e.g., memory, time estimation, decision-making).

Finally, recent research focused on how humans extract abstract knowledge, and generalize this knowledge to other situations [90,162–164]. Indeed, abstracting the action sequence structure of the Thunderstruck song may be useful for future learning. Transferring the abstract structure of the Thunderstruck song when learning a novel song that shares a similar structure should improve learning [165].

## Methods

Below we provide a full description of the ACDC model; parameter values for all simulations are reported in Table 1, and code is available from https://github.com/CristianBucCalderon/ACDC.

### The associative cluster-dependent chain (ACDC) model for flexible motor timing

Our ACDC model contains four main modules (Fig 1): an input layer (Fig 1A), an RNN (representing premotor cortex; Fig 1B) and a BG-thalamus unit (Fig 1C).

The input layer (Fig 1A) consists of a vector of neurons, of which a subset is activated, representing sensory or other context that would signal the identity of the sequence to be produced or learned.

Crucially, the dynamics within the ACDC model evolve as a sequential unfolding of RNN-BG-thalamus-RNN (i.e., cortico-basal ganglia) loops, depicted by the light blue arrows in Fig 1. The sequence starts with the activation of a cluster (i.e., densely interconnected) of excitatory RNN neurons (Fig 1B). Each cluster will come to encode the $i^{th}$ element in the action sequence. As opposed to single unit, clustered neurons provide a biologically plausible mechanism for supporting persistent activation within the cluster given a phasic input (i.e., an attractor, [81,166]). In prefrontal cortical–BG models, such clusters are referred to as "stripes" based on their anatomical existence, and are independently gated by BG [167]. Once a cluster is activated, the RNN temporarily settles on an attractor state indicating the ordinal position (order or rank) in the sequence, analogous to how distinct PFC stripes code for ordinal

**Table 1. Parameter values for all simulations.**

| Parameters | Values |
|---|---|
| $\alpha_{1\ (RNN)}$ / $\alpha_{1\ (RNN\text{-}Go)}$ | 0.01 / 0.00002 |
| $\alpha_{2\ (RNN)}$ / $\alpha_{2\ (RNN\text{-}Go)}$ | 0.1 / 0.4 |
| $W_{max\ (RNN)}$ / $W_{max\ (RNN\text{-}Go)}$ | 1 / 0.05 |
| $b$ | 0.5 |
| $\tau_w$ | 2 |
| $\eta$ | 0.4 |
| $\phi$ | 0.01 |
| $J^{IE}$ | 0.1 |
| $\tau_a$, $\tau_n$ | 10 |
| $\gamma_E$ / $\gamma_I$ | 21.4 / 21 |
| $\tau_{rnn}$, $x^{in}$, $J^{EI}$, $J^{EA}$, $J^{IA}$, $J^{GN}$, $J^{NA}$ | 1 |
| $\lambda_{rnn}$ / $\lambda_a$ | 10 / 10000 |
| $\tau_g$ | 1000 |

positions in phonological loop tasks [167]. However, in ACDC such clusters emerge naturally via learning rather than hard-coded anatomical entities. Moreover, attractor states are maintained via a specific ratio of excitatory to inhibitory inputs: each excitatory neuron projects to a common single inhibitory neuron (orange circle in Fig 1B) which reciprocally inhibits all excitatory RNN neurons. As long as the ratio of excitatory to inhibitory inputs is not perturbed by another input (see below), activation in the cluster will persist and the RNN will continue representing the $i^{th}$ order in the sequence.

In turn, each excitatory RNN cluster projects to its corresponding "Go" unit in the BG (blue arrow 1 from $i^{th}$ cluster in Fig 1B to G node in Fig 1C), and each Go cell accumulates evidence for the $j^{th}$ action associated to the $i^{th}$ order (see [134,168] for related computational models of evidence accumulation in these units, and [169] for empirical data). Striatal Go cells, via the basal ganglia direct pathway machinery [170,171], facilitate response execution by projecting towards the corresponding motor thalamus neurons, from here on termed Action nodes for simplicity (blue arrow 2 from Go to Action nodes in Fig 1C). The BG component summarizes the contributions of more detailed BG circuitry [140,167,172]. In these models striatal neurons accumulate evidence, which via the direct and indirect pathways leads to categorically discrete signals in BG output nuclei, and to disinhibition of the thalamus (e.g., [132,168]). These patterns are also observed empirically in terms of striatum accumulation signals and discrete downstream responses in BG output nuclei once a threshold of accumulation is reached [169]. Here, we lumped together the double inhibition from striatum to Globus Pallidus (GP) and from GP to the thalamus into a single excitatory projection to keep the model simple and tractable. Interestingly, optogenetic stimulation of the GP has been shown to increase the firing rate of motor thalamus neurons [173].

Action nodes possess a negative bias, which acts as a decision threshold, i.e., the net input needs to exceed this bias in order for action to be executed. This feature again summarizes the computational role of the output of the BG, which serves to inhibit action execution until sufficient evidence reaches the threshold for action gating ([132,134]; see also [174]). Therefore, the weight values between Go and Action nodes control the speed of action execution: the BG encode the rhythm. Action execution can be expressed either as a transient or persistent response (see simulations; [23]).

In turn, Action nodes project excitatory connections to three distinct parts of the network simultaneously. First, Action nodes project to the cluster of excitatory neurons in the RNN representing the $i+1^{th}$ order in the sequences (blue arrow 3a in Fig 1). Second, Action nodes project to the inhibitory shared neuron (blue arrow 3b to orange node in Fig 1), that in turn globally inhibits all the clusters in the RNN. In this manner, thalamic Action nodes can update the cortical representation by separately projecting to both inhibitory and excitatory neurons [52,175], enabling the RNN to transition from the current state to the next. That is, the activation of action nodes perturbs the ratio of excitatory to inhibitory RNN inputs in a way that allows the $i^{th}$ cluster to shut down and the $i+1^{th}$ cluster to be expressed. Third, Action nodes project excitatory connections back to their corresponding No Go cells (blue arrow 3c from $j^{th}$ Action node in the thalamus to $j^{th}$ No Go node in the BG, see Fig 1C). In turn, No Go cells strongly inhibit their corresponding Go cells [132,176,177], thereby shutting down evidence in favor of the $j^{th}$ action, and hence stopping the execution of the $j^{th}$ action. This loop is then reproduced with the $i+1^{th}$ RNN cluster and $j+1^{th}$ G-A-N triplet in the BG-thalamus unit, and so forth until the action sequence is performed in its entirety.

Several features of the model should be highlighted. First, each cluster activation within the RNN acts as an attractor state representing the $i^{th}$ element in the sequence. Interestingly, cells in the monkey PMC code for the position in sequence, regardless of the actual movement produced during that position [178–184]. We therefore assume that the neurons forming each

cluster represent rank-order-selective neurons whose activation unfolds sequentially: the RNN encodes order information.

Second, the speed at which each action is executed is driven by how quickly the evidence in the Go nodes of the BG can cross the decision threshold in the Action nodes: the BG encode time information. Indeed, several studies suggest that temporal processing is subtended by the BG in the (non)human primates and rodent brain [1,2,185–189]. Note that there are multiple routes by which timing can be altered within Go nodes in our model: (i) the learned weight value between Go and Action nodes; (ii) a bias input to Go nodes (in addition to that coming from the RNN cluster); and (iii) a multiplicative gain on Go unit activity (see model simulations). As shown in the results section, these separate routes are important for providing timing and rhythm flexibility.

Third, as in many cortico-BG models (e.g., [134,190]), and motivated by anatomical data [191] our model is characterized by topographical organization of actions across the BG circuit and its outputs (i.e., indexed in our model by the subscript $j$ associated in the G-A-N triplet projections). Recent evidence further confirms topographical action representations in BG-thalamocortical loops [192–194], whereby causal activation of specific subregions is related to specific output behaviors [195], and is also supported by human neuroimaging [196] and monkey/rodent neurophysiology studies [1,197–202]. However, in contrast to previous models in which BG gating affords action selection of the corresponding cortical action, in the ACDC model BG gating triggers a cortical dynamical state that initiates the evolution of the *subsequent* item in the sequence.

Fourth, we clarify how the ACDC model combines properties of associative chain and cluster-based models. While the ACDC model does initiate a *chain* via sequential propagation across cortico-BG loops, the timing of such transitions is controlled by learning the weights within the BG-thalamus unit, and moreover, what is learned are transitions between clusters of excitatory RNN neurons representing order in the sequence [27]. Hence, the ACDC model makes use of two distinct conceptualizations of sequence learning, to achieve greater computational flexibility (as demonstrated in the result section).

### Learning in the ACDC model: Hebbian learning for order and Delta rule for time

Learning in the ACDC model takes place in three distinct loci of the network, comprising Hebbian learning for sequence transitions and error-driven learning for precise timing.

First, as previously mentioned, order is coded via persistent activation within clusters of the RNN. However, in contrast to pure associative chain models, the ACDC does not assume any feedforward hard-wired structure, but rather learns it. Selective time-dependent inputs to the RNN (i.e., from the input layer and thalamic Action nodes) activate a subset of neurons within the RNN, which get clustered together through dynamic synaptic weights:

$$\frac{dW_{ij}}{dt} = -\alpha_1((1 - x_i)\bar{x}_j) + \alpha_2(x_i\bar{x}_j(W_{\max} - W_{ij})) \tag{1}$$

where $\bar{x}_j$ is presynaptic activity low-pass filtered over a time scale $\tau_w$; $x_i$ is postsynaptic activity; $\alpha_1$ and $\alpha_2$ are learning rate parameters. When $\bar{x}_j$ and $x_i$ are both simultaneously $> 0$, $W_{ij}$ goes to $Wmax$; otherwise $W_{ij}$ decreases (note that we clamp $W_{ij}$ such that $W_{ij} \geq 0$). Note that $\bar{x}_j(t)$ will be non-zero if unit $j$ is active within the time window from $t - \tau_w \to t$. Note that low-pass filtering is not strictly necessary in our version of the model and was implemented to maintain consistency with previous work in the domain [28]. However, its value should be $< 2$ otherwise all RNN units will tend to be connected and action sequence learning will fail.

Second, Eq 1 is also used to learn connections between the RNN and the Go nodes of the BG module; here, pre- and postsynaptic activity refer respectively to RNN excitatory unit activity and Go nodes activity (weight values between RNN units and Go nodes are randomly initialized from a Gaussian distribution with mean = 0.5 / $N$ and s.d. = 0.1 / $N$, where $N$ is the number of RNN excitatory units).

Third, action specific execution time is coded in the weights connecting Go and Action nodes. Here, we describe time learning as a delta rule, whereby an agent receives a supervisory signal explicitly indicating whether a specific action has been produced before (positively signed signal to increase weights) or after (negatively signed signal to decrease weights) the appropriate time, and is described in Eq 2:

$$\Delta W = \eta(t_{observed} - t_{desired}) \tag{2}$$

where the change in weight ($\Delta W$) between the $j^{th}$ Go and Action nodes is driven by the learning rate $\eta$, and the error computed as the difference between the observed and desired response time ($t$) for each action. Weight values between Go and Action nodes are randomly initialized and drawn from a random Gaussian distribution (mean = 2, s.d. = 0.2). Learning of precisely timed sequences is shaped sequentially: the model first learns to produce the first action at the appropriate time (i.e. until the error $< \varphi$ and $\varphi$ is a low value, see Table 1), then the second, and so forth.

## Mathematical description of the model dynamics

The input layer reflects a vector of $N$ = 200 neurons of which a subset (20) is activated and each neuron excites only one neuron in the RNN.

The dynamics within the ACDC model represent the sequential unfolding of RNN-BG-thalamus-RNN (i.e., cortico-basal ganglia) loops, depicted by the light blue arrows in Fig 1. The loop starts with the activation of a cluster of excitatory RNN neurons, and the dynamics of the RNN excitatory neurons are governed by Eq 3:

$$\tau_{rnn}\frac{dx_i}{dt} = -x_i + \Theta\left(\sum_{j=1}^{N} W_{ij}x_j - J^{EI}x_I + J^{EA}(x_A\gamma_E) + x_i^{in}\right) \tag{3}$$

where $x_i$ and $x_j$ represent post- and pre-synaptic RNN unit activity (purple nodes in Fig 1B) and $W_{ij}$ is the recurrent weight matrix. $J^{EI}$ and $J^{EA}$ represent respectively the weights from the shared inhibitory neuron (orange node in Fig 1B) and from the motor thalamus neurons (from here on termed Action nodes for simplicity) to the excitatory RNN units. $x_I$, $x_A$ and $x^{in}$ represent respectively the activity of the shared inhibitory neuron, Action nodes (see below), and the input to the excitatory RNN units. $\gamma_E$ is the gain on Action nodes activation projected to the excitatory RNN neurons (see below for the functional property of this parameter). $\Theta$, the non-linear transformation function, is governed by $\Theta(x) = (2 / (1 + e^{-\lambda x}))- 1$ (where $\lambda$ is the gain parameter and with additional non-linearity at zero, i.e. $\Theta(x) = 0$ if $\Theta(x) < 0$); and $\tau_{rnn}$ is the encoding constant. Note that input projections and all Action nodes to RNN projections are orthogonal (i.e. some RNN excitatory neurons receive inputs from the input layer, whereas others receive from inputs from Action nodes; each projection excites 20 RNN units). The shared inhibitory $x_I$ activation is described by Eq 4:

$$\tau_{rnn}\frac{dx_I}{dt} = -x_I + J^{IE}x_i + J^{IA}(x_A\gamma_I) \tag{4}$$

where $J^{IE}$, $J^{IA}$ and $\gamma_I$ respectively represent the weights from the excitatory RNN neurons to

their shared inhibitory neuron, the weights from the Action nodes to the shared inhibitory neuron, and the gain on Action nodes activation for the projections towards the inhibitory neuron in the RNN.

In turn, each excitatory RNN cluster projects to its corresponding "Go" cell in the BG (blue arrow 1 from Fig 1B to Go node in Fig 1C), and each Go cell accumulates evidence for the $j^{th}$ action associated to the $i^{th}$ order, following Eq 5:

$$\tau_g \frac{dg_j}{dt} = -g_j + \sum_i^N W_{ij} x_i - J^{GN} n_j \tag{5}$$

where $g_j$ is the activation of the $j^{th}$ Go units, $W_{ij}$ is the weight matrix representing connectivity between RNN and Go units, $x_i$ is the acitivity of the RNN excitatory units, $J^{GN}$ is the inhibitory weight between the $j^{th}$ No Go and Go nodes, $n_j$ is the activation of the $j^{th}$ No Go node, and $\tau_g$ is the encoding constant (with $\tau_g >>> 0$, thereby simulating evidence accumulation-like dynamics). Non-linearity at zero is also applied to Go-units.

Striatal Go cells facilitate response execution by projecting towards the corresponding Action nodes (blue arrow 2 from the Go to Action nodes in Fig 1C), whose dynamics are governed by Eq 6:

$$\tau_a \frac{da_j}{dt} = -a_j + \Theta \left( J^{AG} g_j - b \right) \tag{6}$$

where $a_j$ is the activation of the $j^{th}$ action, $g_j$ is the activation of the $j^{th}$ Go unit, $b$ is the negative bias (i.e. threshold), $\Theta$ is a nonlinear function as in Eq 3, and $\tau_a$ is the encoding constant. $J^{AG}$ is the weight from the $j^{th}$ Go unit to the $j^{th}$ Action unit, and was initially (i.e. before learning) randomly drawn from a Gaussian distribution with mean = 2 and s.d. = 0.2. In turn, Action nodes project excitatory connections to three distinct parts of the network simultaneously. First, Action nodes project to the cluster of excitatory neurons in the RNN representing the $i+1^{th}$ order in the sequences (blue arrow 3a in Fig 1). Second, Action nodes project to the inhibitory shared neuron (blue arrow 3b to orange node in Fig 1), that in turn globally inhibits all the clusters in the RNN. Note that the gain parameter values on Action nodes activity are larger for projections to the excitatory clusters vs inhibitory neuron of the RNN (i.e. $\gamma_E > \gamma_I$). This allows the activation of Action nodes to perturb the ratio of excitatory to inhibitory RNN input in a way that allows the $i^{th}$ cluster to shut down and the $i+1^{th}$ cluster to be expressed. Third, Action nodes project excitatory connections back to their corresponding No Go cells (blue arrow 3c from $j^{th}$ Action node in the thalamus to $j^{th}$ No Go node in the BG, see Fig 1C). The dynamics of No Go cells are in turn dictated by Eq 7:

$$\tau_n \frac{dn_j}{dt} = -n_j + J^{NA} a_j \tag{7}$$

where $n_j$ is the activation of the $j^{th}$ No Go node, $J^{NA}$ is the weight from the $j^{th}$ Action unit to the $j^{th}$ No Go unit, $a_j$ is the activation of the $j^{th}$ Action node, and $\tau_n$ is the encoding constant.

In Table 1 we report the parameter values used for all 9 simulations described in the main text.

## Supporting information

**S1 Audio file. Simulation 7: Thunderstruck song as reproduced by the ACDC model.** This audio file shows the ability of the ACDC model to learn the second guitar riff of Thunderstruck

reproduce the 16 actions (associated to 6 notes, see main text) in the correct order and tempo.
(MP4)

**S2 Audio file. Simulation 7: Thunderstruck song following a bossa nova rhythm.** This audio file shows the ability of the ACDC model to perform temporal compositionality. The ACDC model can produce flexibly produce the previously learnt guitar riff (S1 Audio file) following a bossa nova rhythm without any further training.
(MP4)

**S1 Video. Simulation 7: Dynamical visualization of RNN and action nodes activity coupled with simulation-based Thunderstruck song sound.** The top left panel shows how RNN sequential and persistent activity unfolds as a function of time. The bottom left panel is a visualization of RNN dynamics as a neural trajectory in principal component (PC) space. The neural trajectory displays a pattern of sequential attractor states. The right panel displays how activity in each Action node (and hence Thunderstruck song note) is executed at the learned action time.
(MP4)

**S2 Video. Simulation 9: Dynamical visualization of RNN and Action nodes activity.** The left panel shows how activity in each Action node is executed at the learned action time, each color represents the activation of a specific A node in the thalamus. Given the structure and mechanism described in Fig 1, the right panel displays the neural RNN trajectory showing that each action execution triggers a switch from the $i^{th}$ to the $i^{th+1}$ attractor state.
(MP4)

## Acknowledgments

We thank the members of the Frank and Verguts lab for helpful discussions, and Jose Miguel Buc Chavez for the bossa nova rhythm description.

## Author Contributions

**Conceptualization:** Cristian Buc Calderon, Tom Verguts, Michael J. Frank.

**Formal analysis:** Cristian Buc Calderon.

**Supervision:** Tom Verguts, Michael J. Frank.

**Writing – original draft:** Cristian Buc Calderon.

**Writing – review & editing:** Cristian Buc Calderon, Tom Verguts, Michael J. Frank.

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
