## [Decision Letter · Decision Letter 0]

23 Nov 2021

Dear Dr. Calderon,

Thank you very much for submitting your manuscript "Thunderstruck: The ACDC model of flexible sequences and rhythms in recurrent neural circuits" for consideration at PLOS Computational Biology. I apologize for the delay; we had to wait unusually long to receive all the reviews.

As with all papers reviewed by the journal, your manuscript was reviewed by members of the editorial board and by several independent reviewers. In light of the reviews (below this email), we would like to invite the resubmission of a significantly-revised version that takes into account the reviewers' comments. As you'll see from the comments below, there were two major issues that should be addressed. One concerns comparison with other models, in particular clarifying what is really new here and what can/can't be done by other models under particular (e.g., biological) constraints. The second issues concerns clearing up some uncertainty about exactly how the model works and how it produces the results.

We cannot make any decision about publication until we have seen the revised manuscript and your response to the reviewers' comments. Your revised manuscript is also likely to be sent to reviewers for further evaluation.

Sincerely,

Samuel J. Gershman

Deputy Editor

PLOS Computational Biology

Reviewer's Responses to Questions

**Comments to the Authors:**

Reviewer #1: **********************************************

Major Points ****************

* This article describes a model where a cortical module, a basal ganglia module, and a thalamic module interact to perform action selection at adjustable timings. Specifically, the model combines a chain-like structure that is distributed over the three modules and that encodes the sequence order, while the modulation of action timing is implemented through dedicated reciprocal thalamo-basal connections. The larger structure of the network is hard-wired, while some adjustable synaptic weights are learned through local learning rules and simple scalar reinforcement signals.

* The biggest strength of the paper is its very engaging narrative, and the large effort put into showcasing the functionality of the network with nice detailed figures, videos, and code.

* I am not confident about the actual extent of the relative computational advantages of the proposed model - but I don't think that this should preclude publication in Plos CB as this model does have some specific design elements that do work.

-First, state-space models are actually able to do temporal rescaling (Flexible timing by temporal scaling of cortical responses, Wang et al. Nat. Neuro 2018) and can be trained to behave well in a variety of settings using continuous reinforcement signals (Recurrent neural networks as versatile tools of neuroscience research, Barak, Curr Opin Neurobiol, 2017), including by local learning rules (Predicting non-linear dynamics by stable local learning in a recurrent spiking neural network, Gilra and Gerstner Elife 2017; Local online learning in recurrent networks with random feedback, Murray, Elife 2019) that can sculpt a diversity of connectivity structures (Emergence of functional and structural properties of the head direction system by optimization of recurrent neural networks, Cueva et al, ICLR 2020). Therefore, the relative advantage of the currently suggested model lies in its ability to flexibly perform temporal operations while only being trained with simpler learning rules.

-Second, it looks like ring-like models (e.g. Recurrent networks with short term synaptic depression, York and Van Rossum, J Comput Neurosci 2009; Learning multiple variable-speed sequences in striatum via cortical tutoring, Murray and Escola, Elife 2017) could perform very similar operations as the proposed network. Indeed, such ring-like models can be delayed by not providing excitation to the circuit (resulting in a silent network). In addition, in such models, larger background currents lead to faster dynamics, so these models support temporal rescaling. Finally, 'temporal compositionality' (i.e. following a higher-level rhythm signals) could be achieved by using a 'rhythm command' consisting of a time-varying background input (using larger inputs when wanting a fast interval, and smaller inputs when wanting to slow down the rhythm). The relative advantage of the currently proposed model is therefore to suggest how to achieve these types of operations using a larger cortical-basal-ganglia-thalamic network, and using a 'rhythm command' that consists of timed pulses (as shown in fig. 7b) instead of the above mentioned time-varying background input.

* Some comparisons are made with electrophysiology and behavior, and the authors demonstrate that some basic features of neuronal activity are consistent with the model.

-One point that I think should be clarified is that the model focuses on action *selection* rather than action execution. Indeed, even for simple actions such as 'playing the piano' as suggested by the authors, the muscle commands that need to be generated by the brain are continuous (A neural network that finds a naturalistic solution for the production of muscle activity, Sussillo et al, Nature Neuroscience 2015; Motor Cortex Embeds Muscle-like Commands in an Untangled Population Response, Russo et al., Neuron 2018; Thalamic control of cortical dynamics in a model of flexible motor sequencing, Logiaco et al., Cell Reports 2021).

-It would be nice to mention that the involvement of basal ganglia in *fixed order* sequences of movement is a debated topic (Motor learning, Krakauer et al., Compr. Physiol. 2019, available here: https://www.researchgate.net/profile/John-Krakauer/publication/331774116_Motor_Learning/links/5c90db8b45851564fae71871/Motor-Learning.pdf).

-It could be nice to mention that some evidence suggests that neurons at the output nuclei of basal ganglia have a more categorical type of response rather than an accumulation type of response as suggested by the model (Basal ganglia subcircuits distinctively encode the parsing and concatenation of action sequences, Jin et al., Nature 2014).

-It would be nice to discuss that the direct connections from basal ganglia to thalamus are thought to be mostly inhibitory - but disinhibition is possible (through the thalamic reticular nucleus or more directly, e.g. Inhibitory Basal Ganglia Inputs Induce Excitatory Motor Signals in the Thalamus, Kim et al., Neuron 2017).

Minor Points ****************

* The temporal rescaling operation also induces a shift.

* Just by reading the main text, I was not sure whether the plastic connections in all parts of the network (recurrent RNN connections, RNN -> go connections; go -> 'action' connections) were learned simultaneously. I think this should be clarified. Also, in Fig. 1, it would be nice to highlight (with a specific color) the plastic connections, and to put symbols to clearly identify the corresponding types of learning rule used.

* The discussion about the behavioral evidence of learning sequences one item at a time, now present in the Methods, would in my opinion be very useful to include in the main text.

* In Fig. 2, it would be nice to

-mention explicitly the meaning of the indices (ith cluster, jth ... ). Could it be possible to use indices that correspond to 'action selection order'?

-in panel A, it would be nice to always use different shades of color to indicate the progression over learning

* lines 735-736: for J^{AG}, are you describing the initial weights (before learning)?

Reviewer #2: Summary:

This work proposes a multi-regional model, the associative cluster-dependent chain (ACDC), that learns to flexibly execute actions. The flexibility consists in being able to temporally shift multiple actions in time, recombine them, and scale their duration, even though the network was not explicitly trained on all the possible actions in a supervised manner. The authors also show that with their model they are able to account for behavioral and neurophysiological observations. They then make use of their model’s flexibility to replay a learned song in a different rhythm. The novelty in this work consists in proposing a biologically plausible model with a wide-ranging amount of flexibility that was not previously seen in a single model. It is also interesting in that it generates some insight into possible computational mechanisms that could give rise to a couple of behavioral and neurophysiological empirical observations.

However, to fully appreciate the contributions of this work and understand how exactly it compares to and extends previous work, the authors need to enhance clarity in the following general ways. Currently, it is unclear (1) what exactly is the novelty in this work, (2) how the model is trained and works, and (3) to what extent the authors claims are supported by the actual findings (Figures).

It is unclear what exactly the novelty in this work is... If it is the biological plausibility, then the authors should carefully spell out what aspects of the model are in fact biologically plausible (e.g. in Introduction). While the overall modular architecture seems to be biologically plausible, some of the training mechanisms are not necessarily so (e.g. supervised training in certain parts of the network; pre-established action modules).

With regards to other work, the authors argue that state-space models, for instance, draw on non-biological learning mechanisms and cannot encode multiple sequences nor exhibit temporal scaling (p.5). But more biologically realistic learning rules have been proposed that are able to achieve good performance such as “feedback alignment” (Lilicarp et al, 2016) and “information alignment” (Kunin et al 2020). It is generally possible to encode multiple sequences in these models, and temporal scaling may be achieved easily simply by training on a diverse repertoire of sequences. Also, recent work suggests that with a reservoir of relevant dynamics it may be enough to train output weights only to achieve flexibility (https://arxiv.org/abs/2105.14108). These frameworks are quite flexible in the tasks that can be trained/executed, while the model the authors propose appears to be limited to executing particular action sequences; it is unclear how other kinds of tasks can be achieved in this framework. The authors should therefore discuss the tradeoffs in “baking in” certain prios via the proposed architecture.

If the novelty rather consists in achieving a wide-ranging amount of flexibility through a single model, then the authors should spell this out clearly and discuss tradeoffs/how they integrated different aspects.

The authors may also benefit from comparing to the following works: https://www.sciencedirect.com/science/article/pii/S0893608020303312?dgcid=rss_sd_all and https://www.nature.com/articles/s41593-019-0415-2

It is unclear how exactly the model is trained, where outputs are read out from and how, and which aspects of the model yield the purported flexibility.

The authors argue at one point, for instance, that it is the “modularity” that “affords independent (and flexible) manipulation of sequence order and action timing” (p.5 bottom). But how is this conclusion supported by the reported results/figures? It is in fact unclear from the figures and explanations how the model is trained, how it executes actions (how actions are read out), and how/based on which aspects the flexibility emerges (more specific points below).

In many cases it seems like there is quite a wide gap between explanations/interpretations and what is actually depicted in the figures (specifics below). The manuscript would greatly benefit from updated Figure legends and explanations that carefully describe what is shown in the figures and how they support particular claims.

Specifics:

p.2: “We argue that this limitation emerges from the fact that oder information and timing are typically stored in the same neural network weights” - how is this supported by your findings/what are the tradeoffs?

p.5: Comparison to state-space models - as explained in (1) this paragraph would benefit from a more careful evaluation and discussion of tradeoffs.

p.5: “However, the mechanism for such compositionality in neural networks remains unknown” - This seems to suggest the main reason/novelty in this work consists in demonstrating how ‘compositionality’ is achievable. If so, then the manuscript should be restructured around this point and tradeoffs should be clearly explained (as mentioned in (1) - how many different types of tasks can be composed in this framework?). However, it seems that the type of flexibility achieved goes beyond just compositionality.

p.5: “The modularity thereby affords independent (and flexible) manipulation of sequence order and action timing” - how is this supported by findings? What about previous work with modular networks?

p.6 Fig1: There is no information/explanation on biological plausibility of this architecture/training regime etc.

p. 7, l.183-185/Fig2: How are actions trained? What are the inputs? Which part of the network are actions read out from? More generally, how does learning proceed throughout the entire model - are the various component trained jointly or sequentially? What are the inputs/outputs of the various components?

p.8 Fig2: It is unclear how weights in D and E support the claim that there are “clusters” in the network. In F, it is unclear what the inputs are to the G and N nodes and how these are trained.

p.9 l.220: “We show that a previously learnt action sequence with temporal asynchrony can be flexibly reproduced” - based on Fig2A it seems that asynchrony is explicitly learned, so it would not be surprising that this will work (in the same way that supervised networks can achieve flexibility via training set). More detail on how exactly training proceeded/what actions were fed in as inputs would really help here.

p.11 Fig3: Generally, it would be good to understand where these action sequences are read out from, what the inputs are, and what the other parts of the network are doing when throughout these sequences. It is unclear how D suggests ‘Compression’ as some of the actions seems to be aligned with the ones in B, while others seem slightly temporally shifted. Similarly, it is unclear how the right panel shows ‘Dilation’ as some of the actions appear in the exact same position as in the right panel of B and others only seem slightly shifted. Re F, it is unclear how this supports ‘Compositionality’ since the order is preserved and only the duration of particular actions seems extended.

p.12 l 292/Fig 4: “via a learning mechanism” - what was the learning mechanism/how were these trained? How was multiplicative signal fed in, how were weights manipulated?

p.13/Fig5: What was done to achieve the transformation from A to C? As in what combination of inputs/manipulations/training was necessary to achieve this?

p.15/Fig7: How does part A show clusters in the network? How are switches “triggered” based on what we see in A? How are panels A and B related - are activations from the RNN fed into the G units? If so how are they wired, which part initiates an action? Perhaps this figure should be integrated with Fig2 for clarity.

p.16, l.386: Based on the video and how activity evolved in PC space, it is not clear that these are in fact attractors - a fixed point analysis should accompany these analyses to support this conclusion. What initiates the switch from one state into another, assuming it is input driven? Also, the video shows 6 actions but only 5 distinct states in PC space - it seems some actions are mapped on top of each other, while others are not.

**Have the authors made all data and (if applicable) computational code underlying the findings in their manuscript fully available?**

Reviewer #1: Yes

Reviewer #2: Yes

PLOS authors have the option to publish the peer review history of their article (what does this mean?). If published, this will include your full peer review and any attached files.

Reviewer #1: **Yes: **Laureline Logiaco

Reviewer #2: No
---

## [Decision Letter · Decision Letter 1]

21 Jan 2022

Dear Dr. Calderon,

We are pleased to inform you that your manuscript 'Thunderstruck: The ACDC model of flexible sequences and rhythms in recurrent neural circuits' has been provisionally accepted for publication in PLOS Computational Biology.

Before your manuscript can be formally accepted you will need to complete some formatting changes, which you will receive in a follow up email. A member of our team will be in touch with a set of requests. Also please address in your final version the minor requests made by the reviewer below.

Best regards,

Samuel J. Gershman

Deputy Editor

PLOS Computational Biology

Reviewer's Responses to Questions

**Comments to the Authors:**

Reviewer #1: The authors have accounted for my comments, and have greatly clarified the text. I think this manuscript is ready for publication. I have three very minor formal suggestions for the final version of this paper:

1) I think that, on line 587, the authors use a subscript '3' but actually meant to refer to '(footnote 3)', not the third reference.

2) In the caption Fig. 2E, when using the bold subtitle 'RNN ith cluster learns to project to jth Go node.', I think it would help to add '(see Fig. 1, 'Detailed architecture', for the meaning of the indices)'

3) In Fig. 2A, for the two top left graphs (dark blue and red), I feel that the saturation of the color shades towards black makes it harder to understand the panel. I think it would be helpful to either use a color scheme where the various shades of the color are visible even for the trials that represent early learning (the best option in my opinion - and it does not matter if there is little difference in coloring for the overlaid trials); or to explicitly say in the legend that the 'black trials' are actually early trials where the different shades are not distinguishable.

**Have the authors made all data and (if applicable) computational code underlying the findings in their manuscript fully available?**

Reviewer #1: Yes

PLOS authors have the option to publish the peer review history of their article (what does this mean?). If published, this will include your full peer review and any attached files.

Reviewer #1: **Yes: **Laureline Logiaco

---

## [Editor Report · Acceptance letter]

27 Jan 2022

PCOMPBIOL-D-21-01530R1 

Thunderstruck: The ACDC model of flexible sequences and rhythms in recurrent neural circuits

Dear Dr Calderon,

I am pleased to inform you that your manuscript has been formally accepted for publication in PLOS Computational Biology. Your manuscript is now with our production department and you will be notified of the publication date in due course.

With kind regards,

Zsofia Freund
